# Estimating Scalp Moisture in a Hat Using Wearable Sensors

**DOI:** 10.3390/s23104965

**Published:** 2023-05-22

**Authors:** Haomin Mao, Shuhei Tsuchida, Tsutomu Terada, Masahiko Tsukamoto

**Affiliations:** 1Graduate School of Engineering, Kobe University, 1-1 Rokkodai-cho, Nada-ku, Hyogo, Kobe 657-8501, Japan; maohaomin@stu.kobe-u.ac.jp (H.M.); tuka@kobe-u.ac.jp (M.T.); 2Education and Research Department Center for Interdisciplinary AI and Data Science, Ochanomizu University, 2-1-1 Otsuka, Bunkyo-ku, Tokyo 112-8610, Japan; tsuchida.shuhei@ocha.ac.jp

**Keywords:** scalp care, scalp moisture content, wearable sensor, machine learning

## Abstract

Hair quality is easily affected by the scalp moisture content, and hair loss and dandruff will occur when the scalp surface becomes dry. Therefore, it is essential to monitor scalp moisture content constantly. In this study, we developed a hat-shaped device equipped with wearable sensors that can continuously collect scalp data in daily life for estimating scalp moisture with machine learning. We established four machine learning models, two based on learning with non-time-series data and two based on learning with time-series data collected by the hat-shaped device. Learning data were obtained in a specially designed space with a controlled environmental temperature and humidity. The inter-subject evaluation showed a Mean Absolute Error (MAE) of 8.50 using Support Vector Machine (SVM) with 5-fold cross-validation with 15 subjects. Moreover, the intra-subject evaluation showed an average MAE of 3.29 in all subjects using Random Forest (RF). The achievement of this study is using a hat-shaped device with cheap wearable sensors attached to estimate scalp moisture content, which avoids the purchase of a high-priced moisture meter or a professional scalp analyzer for individuals.

## 1. Introduction

Hair and scalp health are essential parts of human health. Because the scalp is more delicate than the other skin on the human body, improper or no scalp care may lead to health issues, varying from hair loss to diseases according to severity. Practically, most people only pay attention to the protection of their hair. When people choose shampoo, conditioner, and other hair care products, they tend to focus on whether these products are suitable for their hair rather than whether these products are suitable for their scalp. Due to the interdependence between the scalp and hair, the hair blocks the ultraviolet radiation that is harmful to the scalp and helps the scalp moisturize, and a healthy scalp can breed beneficial hair fibers [1]. Once problems such as baldness and scalp damage occur, these problems are more difficult to treat than hair damage; thus, scalp care is most important.

Although some scalp care products are obviously for avoiding scalp health problems, it is difficult to say whether these products are effective because few have corresponding medical literature proof [2]. An unbalanced diet and disordered lifestyle could also cause scalp health conditions to change constantly. People cannot solve such a problem with scalp care products. However, we could roughly know the scalp condition by monitoring data, such as scalp temperature and moisture content, and because human hair and scalp are easily affected by environmental temperature and humidity, it is also necessary to monitor environmental data.

First, optimal scalp temperature is generally considered to be approximately 34 °C [3]. The moisture content of the skin’s stratum corneum is roughly 10–30% [4], and the scalp moisture is higher than that, for example, 35–50%. If the scalp temperature is lower than that, blood circulation to the scalp is reduced, insufficient oxygen is delivered, toxins accumulate around the scalp, and hair growth is impeded [5]. If the scalp temperature is too high, itching and pain will occur, and the scalp may obtain a parasitic infestation and develop inflammation [6]. Hair loss and dry scalp will occur if scalp moisture is less than 10% [7]. Moreover, scalp moisture is easily affected by environmental humidity [8]. The most suitable environmental humidity for the skin is around 60%, which is the same for the scalp. High-humidity environments mean that mold and bacteria are likely to grow, sebum secretion becomes active, dandruff is produced, and the scalp takes on a bad odor. In addition, when the environmental humidity falls below 40%, static electricity will be generated and damage the scalp surface. Although a temperature sensor or thermistor could be used to sense scalp temperature continuously, there are no simple sensors or devices for sensing scalp moisture accurately, and specialized equipment is expensive and unsuitable for routine measurement because of its large size.

Therefore, in this study, we developed a hat-shaped device equipped with wearable and environmental sensors to estimate scalp moisture content in a hat environment. We used the hat-shaped device in a specially designed space to collect scalp data and a mobile moisture meter to collect ground truth of scalp moisture under different environmental conditions to train four machine learning models: Support Vector Machine (SVM), Random Forest (RF), Neural Network (NN) with one-dimensional convolution layers (Conv1d), and NN with Gate Recurrent Unit (GRU). SVM, RF, and NN are typical machine learning models. Machine learning models have usually been programmed to recognize a specific pattern, which cannot be directly observed by people, after being trained by a data set. Machine learning models can learn some features from the data set and make inferences based on their algorithm, which means machine learning models can estimate previously unobserved data after training. The estimation results of scalp moisture were also evaluated under inter-subject and intra-subject. Finally, we experimented with estimating scalp moisture in a daily environment and obtained an application method with estimation results.

## 2. Related Research

### 2.1. Human Thermal Model

Human thermal models describe the heat exchange and balance of the human body. Still, the complexity of each model is quite different. Among these models, Gagge et al. first proposed the 2-node model in 1971, which assumed a human body as a two-layer structure of skin layer and deep layer and gives biological data generated from heat exchange, such as body temperature and sweat rate, as a mathematical expression [9]. The general structure of the 2-node model is shown in Figure 1. The 2-node model’s segmentation of human body parts is straightforward; however, in human thermal models developed subsequently, the number of human body segments has increased, and the expression of heat exchange in various parts of the human body has become more detailed. For example, the 25-node model proposed by Stolwijk et al. divides the human body into six parts and four layers [10]. Tanabe et al. proposed the 65MN model based on the 25-node model, and Smith et al. proposed Smith’s model, which divides the human body into 15 parts [11,12].

Furthermore, some models are called biothermal models, which can be researched for therapies for some diseases or studying the habits of animals and plants if the functional objects of the human thermal models are extended to specific objects. Wakamatsu et al. proposed a treatment for brain hypothermia through a patient’s biothermal model [13]. Men-Chi et al. predicted the transient temperature of diseased wall tissue by building a biothermal model of atherosclerotic patients to help doctors improve treatment procedures [14]. Separately, Luecke et al. studied how California sea lions behaved in water through biothermal models [15]. Romero et al. predicted the growth of four commercial varieties of cane [16].

### 2.2. Measurement of Scalp Moisture Content

As the measurement of scalp moisture is included in the scope of the measurement of skin moisture, there have been few studies on the measurement of scalp moisture alone. Generally, skin moisture measurement is conducted by a moisture sensor. Previously, moisture sensors were developed based on traditional humidity sensors, such as interdigital capacitance [17]. Recently, new moisture sensors for skin moisture measurement have been continuously developed. Lu et al. developed an integrated, flexible, and small sensor system that can be fixed on a human finger to measure moisture and temperature in real time [18]. Mondal et al. developed an anodized aluminum oxide (AAO)-assisted $MoS_{2}$ honeycomb resistive humidity sensor with higher sensitivity [19]. These sensors are difficult to apply to the measurement of scalp moisture because it is difficult to keep the electrode part close to the scalp due to the obstruction of hair, so specialized equipment for scalp moisture must replace electrodes for probes. Some specialized devices can be used to measure scalp moisture. Still, most are expensive and difficult for individuals to purchase and use in daily life, such as transepidermal water transpiration meters or highly sensitive stratum corneum thickness and moisture meters. Thus, this paper proposes an estimation method for scalp moisture to replace the measurement of scalp moisture in daily life.

### 2.3. Wearable Sensors in Clothing

Wearable sensors are ubiquitous for data sensing in the clothing environment. Many studies have shown that placing sensors in clothes, hats, shoes, etc., can monitor the wearer’s behavior patterns and improve their daily life. For example, Farringdon et al. made a jacket that uses fabric stretch sensors to measure upper limb and body activity [20]. Jayasinghe et al. built inertial sensors into various clothes to analyze and classify the wearer’s behavior patterns [21]. Pham et al. put a wireless accelerometer in shoes and built a CNN that uses the accelerometer data to predict the seven kinds of daily activities of wearers [22]. Li et al. put optical fiber-Bragg-grating-based sensors into functional textiles and obtained the wearer’s body temperature in real time through a weighted coefficient model constructed based on the body surface temperature where the textiles are located for health care [23]. Shahnaz et al. developed a smart hat that uses sonar sensors to detect obstacles on a straight path and a three-axis acceleration sensor to detect the behavior of older people wearing the smart hat to prevent them from falling [24]. Chang et al. installed a camera on a hat of a young child to recognize the objects that the child had seen and output it into audio so that the child could learn quickly in daily life [25]. In addition, smart shoes that can generate electricity have been proposed, which can provide new energy solutions [26].

### 2.4. Sensor Technology toward Smart Wearable System

With the complexity of the application scenarios of wearable sensors, the miniaturization of wearable sensors is a trend, which means wearable sensors must be portable and flexible enough nowadays. The developed biosensors for wearable systems have achieved some advancements in recent years. Wang et al. reported a wearable electrochemical biosensor to analyze sweat in physical exercise and at rest [27]. The biosensor has monitored the amino acid levels of the wearer to assess the risk of metabolic syndrome. Ferro et al. created a submillimeter high-power supercapacitor and a biomolecule probe, which provided a great possibility for the miniaturization of biosensors [28]. Wang et al. proposed a flexible and low-cost humidity sensor with fast responses and successfully applied it to detect human breathing and provide an electrical safety warning for bare hands and wet gloves [29]. Yang et al. proposed a wearable sweat sensor that could detect vitamin C and uric acid based on Metal-Organic Frameworks (MOFs). The sweat sensor can be attached to the skin due to good air permeability [30]. Additionally, Nawaz et al. claimed that Organic Electrochemical Transistors (OECTs) are suitable for manufacturing biosensors [31]. Because of electrolyte gating and aqueous stability, OECTs could be operated within a living organism.

## 3. Proposed Method

This study aimed to help people’s scalp care by sensing scalp data in daily life. Thus, we estimated the scalp moisture content, an essential indicator of scalp health, based on machine learning using wearable and environmental sensors attached to a hat. First, we obtained scalp and environmental data in a specially designed space with controlled environmental temperature and humidity. Second, we trained the machine learning models as pre-trained models using the obtained data. When new scalp and environmental data obtained by the hat-shaped device are input to the pre-trained models, scalp moisture content would be estimated as an output.

We selected some features for training the machine learning models by referring to the variables of the 2-node model. In the 2-node model, the heat exchange inside the human body is determined from three kinds of data: individual differences (height, weight, age, and sex), environmental conditions, and biometric information. We did not consider individual differences in this study. Environmental conditions can be regarded as environmental temperature and humidity. Because the two nodes of the 2-node model are the skin and the body core, we adopted the scalp surface and body core temperature as biometric information for estimating scalp moisture. We also added heart rate to the machine learning features to express the activity of the human body. For collecting the data mentioned above, we used different sensors. For aggregating these sensors, we developed a hat-shaped device with wearable and environmental sensors attached, as shown in Figure 2. The hat-shaped device is not only used to collect data but also to estimate scalp moisture content in real time based on a trained machine-learning model.

In the hat-shaped device, we fixed an NTC thermistor for scalp surface temperature measurement and a DHT22 for the internal environmental temperature and humidity measurement in the hat. We mounted an Arduino Nano, Bluetooth module RN-42, mobile battery for power supply, NTC thermistor for core body temperature measurement, pulse sensor for heartbeats measurement, and DHT22 for external environmental temperature and humidity measurement outside the hat. We measured scalp surface temperature, core body temperature, heartbeats, internal hat temperature, internal hat humidity, external hat temperature, and external hat humidity. The corresponding relationship between sensors and data is shown in Table 1.

Moreover, we also developed an Android application mainly to facilitate data acquisition from experimental participants. The application could store the data from the hat-shaped device in a small local database on a mobile phone with SQLite3. It could also control environmental control equipment with REST API and record correct machine learning data. A screenshot of it is shown in Figure 3. The interface of the application is roughly composed of six parts. The top toggle button is used to connect the Bluetooth module RN-42. The four toggle buttons below it are used to control the experimental environment. The following two buttons are used to start and stop the hat-shaped device sensor. Next is the countdown. The editable textbox and button are used to record the ground truth of scalp moisture. The bottom space is for showing the application log massage. The usage of the application will be described in Section 4 with the experimental process.

## 4. Experiment

Because changes in environmental temperature and humidity would affect the water content of the stratum corneum [8], we obtained the biometric data from experimental participants and environmental data around experimental participants to train and evaluate the estimation results of the machine learning models while regulating the current environmental temperature and humidity. We collected 15 participants’ data in a specially designed space.

### 4.1. Experimental Environment

Our experimental environment is shown in Figure 4. The specially designed space is a 1.5 m × 1.5 m × 2.0 m pipe-type booth containing environmental control equipment (heater, cooler, humidifier, and dehumidifier). We used the equipment and a central cooler to modify the environmental temperature and humidity in the booth. SwitchBot hub mini could record the infrared signals. We registered the infrared signals of the remote controllers of the heater, cooler, and humidifier in the SwitchBot hub mini for controlling them on and off via the Android application mentioned in Section 3 with the REST API function. Moreover, we set two SwitchBot bots to the position adjacent to the switch button of the dehumidifier and central cooler to turn them on and off via the Android application. The four toggle buttons below the “CONNECT” button in Figure 3 correspond to the switches of the cooler, heater, humidifier, and dehumidifier. The relationship diagram of the environmental control equipment is shown in Figure 5.

During the experiment, we acquired biometric and environmental data by letting the participants wear the hat-shaped device while working in a seated position in the pipe-type booth. When the participants were wearing the hat-shaped device, we stuck the tip of an NTC thermistor to their scalps. We also inserted and fixed the tip of another NTC thermistor in the participants’ right ear canal and attached a pulse sensor to their little fingers. Then, we acquired scalp surface temperature, core body temperature, and heartbeats while modifying the temperature and humidity in the booth.

### 4.2. Acquisition of Ground Truth of Scalp Moisture

Generally speaking, skin sensors are rarely used to measure scalp moisture, and the measurement is inaccurate because it is vulnerable to disturbances, such as air humidity and sweat. As mentioned in Section 2.2, special measurement instruments are also unsuitable for daily life because of their high cost and large size. In this study, the ground truth of scalp moisture was obtained using the Mobile moisture HP19-M developed by Courage+Khazaka Inc. Naturally, in skin measurement, the relative water content can be obtained using the electrode from a water meter attached to the skin, but in the case of the scalp, the electrode cannot be attached to the scalp because the hair blocks it. However, the moisture in the forehead close to the scalp is roughly the same as in the scalp. Therefore, the upper forehead moisture was measured in the experiment and used as the ground truth for machine learning.

### 4.3. Experimental Procedure

Before starting the experiment, participants were told to operate the environmental control equipment according to the given orders, as shown in Figure 6. There are four orders corresponding to different operation sequences of environmental control equipment. We disrupted the sequence of environmental temperature and humidity changes to obtain more data under more environmental conditions, allowing the machine learning model to cope with a variety of environmental modes. For example, in Operation Order 1, the participant should first measure the data under room temperature and humidity for 15 min (no environmental control equipment working). In the second 15 min, he/she turns on the cooler and dehumidifier for measurement. In the third 15 min, he/she turns off the previous equipment and turns on the heater and humidifier. In the fourth 15 min, he/she turns the humidifier off and the dehumidifier on. In the last 15 min, he/she turns the previous equipment off and turns the cooler and humidifier on until the end of the experiment. The other orders follow roughly the same pattern. The switches of environmental control equipment were all completed by operating the Android application. During each 15 min interval, participants would take five times measurements, each lasting 3 min. When starting the measurement, the participants would press the “START SENSING” button in the Android application. The countdown below the “START SENSING” button would automatically begin, and there would be a vibration prompt when the countdown ends. Participants would then press the “STOP SENSING” button, and the countdown reset. After each measurement, the first author of this paper would use the HP19-M mentioned in Section 4.2 to measure the participant’s forehead moisture three times and ask them to fill in the median value of the three measurements of scalp moisture in the editable textbox in the Android application. Every time the moisture was measured, the first author would gently wipe the participant’s forehead with tissue to prevent the sweat produced in the experiment from affecting the measurement. During the experiment, the first author was in another pipe-type booth next to the participants, instructing them to switch the environmental control equipment. In addition, the operation orders were always displayed on a laptop, which was kept on a table for the participants to check at any time. The ground truth measurement in the experiment is demonstrated in Figure 7.

### 4.4. Machine Learning and Data Preprocessing

We used SVM, RF, NN with Conv1d, and NN with GRU for estimating scalp moisture under 5-fold cross-validation. Both Conv1d and GRU can handle time-series data. All models performed a grid search before estimation. Grid search is a technique for optimizing machine learning model hyper-parameters, and it can significantly increase the model estimation accuracy [32]. When training SVM and RF, we used Synthetic Minority Over-Sampling Technique for Regression with Gaussian Noise (SMOGN) to oversample the learning data to avoid insufficient samples under some environmental conditions [33]. When training the NN with Conv1d and NN with GRU, we used early stopping to inhibit overfitting [34]. Mean Absolute Error (MAE) and Root Mean Square Error (RMSE) were used to evaluate the estimated results. MAE is the mean of the absolute errors of the difference between the ground truth of scalp moisture and estimated scalp moisture values. RMSE is the root of the mean of the square of the difference between the ground truth of scalp moisture and estimated scalp moisture values. The SVM and RF were implemented in Scikit-learn [35], and NN with Conv1d and NN with GRU were implemented in TensorFlow [36]. Moreover, because the magnitude difference between the input and output data of the models might cause significant errors in the estimated scalp moisture, we normalized the input data. Because we used many different sensors, we hope that the influence of physical quantities could be reduced while training the model. Therefore, we chose z-score normalization, which is given by Equation (Equation 1) [37]:(1)$$\begin{array}{c} X_{norm}=\frac{X-\mu }{\sigma } \end{array}$$
where *X* is raw data, μ is the mean of raw data, and σ is the variance of raw data. $X_{norm}$ is normalized data.

Furthermore, we expanded the features of the input sensor data while training the models. Because each feature of sensor data is related to time, we added the mean and variance of the ten sets of data (approximately 20 s) before a certain time point of the same feature value to the current moment data set. The input sensor data set was increased from 7 to 21 dimensions. We obtained a total of 375 sets of data from 15 experimental participants. For each fold cross-validation, 300 sets were used as training data, and 75 sets were used as testing data. When training the machine learning models based on time-series data, because each set of data had a time length of 80 (approximately 160 s), for each fold cross-validation, the shape of 300 × 80 × 21 data was used for training, and the shape of 75 × 80 × 21 data was used for testing.

## 5. Results

We expect that the proposed method has superior estimation ability in daily situations and that machine learning models have general adaptability to different people. We illustrate the estimation results in the inter-subject evaluation in Section 5.1 and the estimation results in the intra-subject evaluation in Section 5.3. In Section 5.2, we evaluate the linear correlation for each feature of learning data to confirm the importance of features. Finally, different from the experiment in Section 4, we conducted another one-day experiment in Section 5.4 to estimate scalp moisture content for one experimental participant in daily life.

### 5.1. Estimation Results of Inter-Subject Evaluation

The average of the 5-fold cross-validation estimation results of the 4 machine learning models with 15 experimental participants is shown in Table 2. The estimated curves of one of the folds are shown in Figure 8.

We can see that from the average 5-fold cross-validation results, SVM has the smallest MAE and RMSE; thus, SVM has the best performance for scalp moisture estimation with multiple experimental participants. RF and NN with GRU also have good estimation results, but the MAE and RMSE of the two models are slightly higher than those of SVM. Moreover, we can see that the MAE and RMSE of NN with Conv1d are oversized; thus, NN with Conv1d has the lowest estimation accuracy.

We thought the estimation results of NN with Conv1d and GRU trained by time-series data should be stable. However, according to Figure 8, we can see that the estimated curves of SVM and RF are much more stable than those of NN with Conv1d and NN with GRU. The RMSE of NN with Conv1d and GRU is higher than those of both SVM and RF. It seems that NN with Conv1d and GRU are more susceptible to outliers. Although we collected the data from 15 participants with 25 samples per person, this may be insufficient to train NN with Conv1d and GRU because they have more deep structures and require more learning data than SVM and RF. Therefore, we plan to increase the number of participants or the number of experiments per participant in the future.

### 5.2. Linear Correlation between Features of Learning Data and Scalp Moisture

Although we are unsure about the relationship between the data collected from the hat-shaped device and scalp moisture, we investigated the linear correlation of the SVM and RF features and the scalp moisture via Principal Component Analysis (PCA), which could explain the importance of features of learning data. PCA could reduce the dimensionality of learning data and minimize information loss of data [38]. Because time-series data cannot be analyzed with PCA, we only show the results of features used in SVM and RF. The covariance coefficients of each feature against the scalp moisture content are shown in Table 3. The features of SVM and RF we chose are the average value of instantaneous, mean, and variance of each sensor data obtained within 3 min. The covariance coefficient ranges from −1 to 1. The judgment of linear correlation is as follows: when the covariance coefficient is between −0.1 and 0.1, there is no linear correlation; and when the covariance coefficient is above 0.5 or below −0.5, there is a strong linear correlation; when the covariance coefficient is between 0.1 and 0.5 or between −0.5 and −0.1, there is a weak linear correlation. We do not discuss positive or negative correlations here.

In Table 3, we can see that the average of instantaneous and mean core body temperature are strongly linearly correlated. Among the remaining averages of instantaneous and mean features, except for external hat humidity, they are weakly linearly correlated. In addition, the variance of most features is uncorrelated. Although the above results are only for the linear situation, we can confirm that the variables we selected regarding the 2-node model are related to scalp moisture.

### 5.3. Estimation Results of Intra-Subject Evaluation

We used the four machine learning models to construct personal models of each participant to conduct the intra-subject evaluation. We still used 5-fold cross-validation for each personal model. Table 4 shows the average MAE and RMSE of the 5-fold cross-validation of 4 models for each experimental participant and the overall average of 15 participants.

Likewise, we can see from Table 4 that the results of SVM and RF are generally better than those of NN with Conv1d and GRU. The average MAE and RMSE of RF of the 15 participants, which are 3.29 and 3.97, are smaller than those of the SVM and of the estimation results in the inter-subject evaluation. Therefore, RF performs better in estimating the individual scalp moisture content, and SVM has good performances in both inter-subject and intra-subject evaluations. The MAE and RMSE of NN with Conv1d are larger than those in Section 5.1, and it is impossible to estimate the scalp moisture correctly. NN with GRU has a smaller MAE for some participants and a larger MAE for the rest. Due to the further reduction in the available data from personal models (only 25 samples per participant), it becomes more challenging to train NN with Conv1d and GRU, so we did not obtain good results with these two models.

The maximum range of moisture content of the scalp is 0–99%, and in general, the difference between the maximum and minimum moisture content of the stratum corneum of the skin should be 15–20% [4]. Although the best MAE result (8.59) obtained in Section 5.1 is slightly larger in the above range, the average result (3.29) of the MAE of RF obtained in this section is acceptable.

### 5.4. Estimation Results in the One-Day Experiment

We conducted a one-day experiment to verify whether the proposed method can estimate the scalp moisture content in daily life. The experiment lasted from 3:00 p.m. until 9:30 p.m. JST. The experimental participant was the first author. The participant wore a hat-shaped device and used the Android application to collect data. The data record interval was 2 s. The ground truth of scalp moisture was measured 3 times every 15 min. The median value of the three scalp moisture measurements was recorded in the Android application. We measured learning data and ground truth in many situations, such as shopping in a supermarket, walking in a seaside park, sitting on a train, stopping at an underground train station, and walking along the street. The one-day itinerary of the experimental participant is roughly as follows. First, he made some preparations at home, went to the supermarket on foot, took the train to the seaside park for a walk, and then took the train home. After returning home, some entertainment activities were carried out. We did not plan the experimental participant’s itinerary, and his behavior in the experiment was arbitrary.

We used the personal model trained in Section 5.3 to estimate scalp moisture content. The estimation results are shown in Table 5, and estimated curves from four models are shown in Figure 9. From Table 5, we can see that the MAE of SVM is the smallest, followed by NN with GRU and RF, and the estimation accuracy of NN with Conv1d is the worst. Although the MAE of SVM is better than the average MAE in Section 5.1, we can see from Figure 9 that the estimated curve of SVM is a straight line, which does not fit the ground truth of scalp moisture very well. The same occurred for RF. On the contrary, although NN with GRU had a significant deviation, the changing trend of its estimated curve fitted the ground truth of scalp moisture very well. Combined with the results in Section 5.3, we believe that if we could acquire more training data, NN with GRU would perform better in estimating scalp moisture in daily life.

To further investigate why the estimation result is not very good, we analyzed the changes in internal hat temperature and humidity, external hat temperature and humidity, and the ground truth of scalp moisture during the one-day experiment; these data are shown in Figure 10 and Figure 11. The two figures show that the temperature and humidity outside the hat change significantly. For example, because the participant was sitting on a train beside a door, every time the train stopped at the platform and the door was opened, there were apparent changes in temperature and humidity. When the participant was walking along the seaside, the relative humidity of the environment was as high as 90%. Other such occurrences in daily life would also lead to changes in environmental temperature and humidity, such as coming out to a balcony and picking up takeout. These situations are challenging to simulate in the experiment of Section 4, so it was difficult for us to collect learning data similar to when the environmental conditions suddenly change for some reason, which is the main factor leading to the inability to estimate scalp moisture accurately in daily life. Therefore, we must expand the simulatable temperature and humidity range in the pipe-type booth and gather more experimental data under different environmental conditions.

## 6. Discussion

### 6.1. Measurement of Core Body Temperature

Theoretically, core body temperature is approximately 37 °C [39]. In the experiment of Section 4, we placed an NTC thermistor in the participants’ ears to obtain core body temperature. Still, the measured value was lower than the standard value because it was difficult to fix the NTC thermistor and reach the eardrum firmly. We think the NTC thermistor only acquired the ear canal’s temperature, not the body’s core temperature. Because it is dangerous for the experimental participants to take action, such as running or walking with NTC thermistors in their ears, safer measurement methods are necessary. Perhaps we might consider a way of fixing the infrared temperature sensor or estimating body core temperature.

### 6.2. Ground Truth of Scalp Moisture

Some ground truth of scalp moisture obtained in the experiments of Section 4 exceeded 90%. This is because we asked the participants to constantly change the temperature and humidity of the environment to obtain learning data under different environmental conditions; some sweated a lot when they switched to a hot environment. Although we wiped the participants’ foreheads before measuring the scalp moisture, it did not completely prevent this from happening. Thus, we think that some of the ground truth of scalp moisture may have deviations, which could lead to a decrease in estimation accuracy.

## 7. Application

Even if we considered that we had not obtained the ideal core body temperature and scalp moisture data in Section 6, some machine learning models performed sufficiently well. Therefore, we present a cloud-service-based method for applying well-performing machine learning models to estimate scalp moisture in real time. The approach is shown in Figure 12. Precisely, we deployed a pre-trained personal machine learning model on a virtual machine in Azure and used MLflow to publish an online service that continuously estimates scalp moisture. MLflow is an open-source platform to manage the machine learning lifecycle, including experimentation, reproducibility, deployment, and a central model registry [40]. The service published by MLflow could exchange learning data and estimation results between the Android application and cloud virtual machine through the REST API. Additionally, the deployed machine learning model could run in a Docker container, and we could mount multiple containers and deploy multiple personal models simultaneously.

## 8. Conclusions

In this paper, we proposed a hat-shaped device equipped with wearable and environmental sensors to estimate scalp moisture content. We estimated the scalp moisture of fifteen experimental participants through four machine-learning models based on scalp surface temperature, core body temperature, internal hat temperature, and humidity, external hat temperature and humidity, and heartbeat obtained from the hat-shaped device. SVM had the smallest MAE in the inter-subject evaluation, and RF had the smallest average MAE in the intra-subject evaluation. In summary, SVM performs well in both inter-subject and intra-subject evaluations. For the one-day experiment, although the MAE of NN with GRU was slightly larger, the trend of the estimated curve was good. Therefore, we need to improve the experimental conditions and obtain more data to estimate scalp moisture content accurately in daily life. In addition, we also presented a cloud-service-based method for applying the scalp moisture estimation results.

Estimating scalp moisture content is feasible by machine learning models based on the learning data obtained from the hat-shaped device. Although some data values are biased, the scalp moisture estimation accuracy will be improved if these biases can be further reduced. As mentioned, one future task of this study is to obtain more data to improve the accuracy of scalp moisture estimation, which still requires many experiments. In addition, in daily life, it is too dangerous to insert the tip of the NTC thermistor into the ear canal to obtain core body temperature. Thus, we must consider other acquisition methods or estimate scalp moisture content without core body temperature.

## Figures and Tables

**Figure 1 sensors-23-04965-f001:**
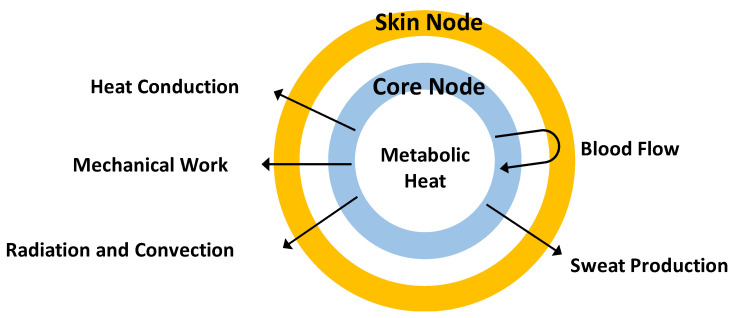
Structure of 2-node Model.

**Figure 2 sensors-23-04965-f002:**
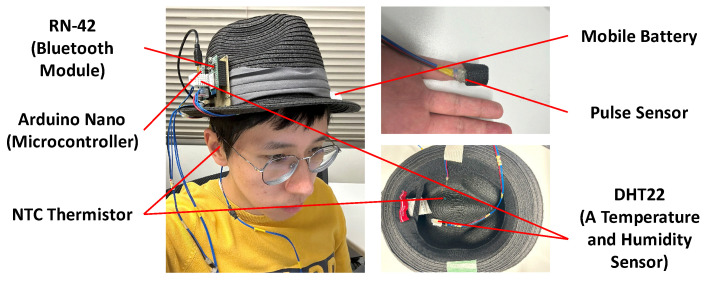
Hat-Shaped Device Implementation.

**Figure 3 sensors-23-04965-f003:**
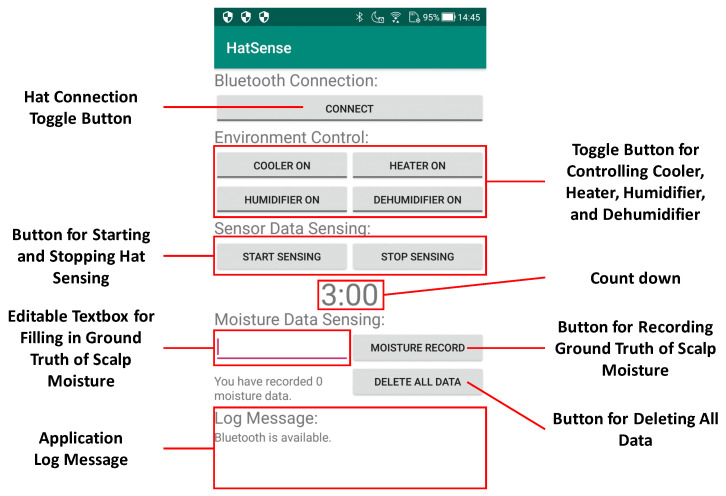
Screenshot of Android Application.

**Figure 4 sensors-23-04965-f004:**
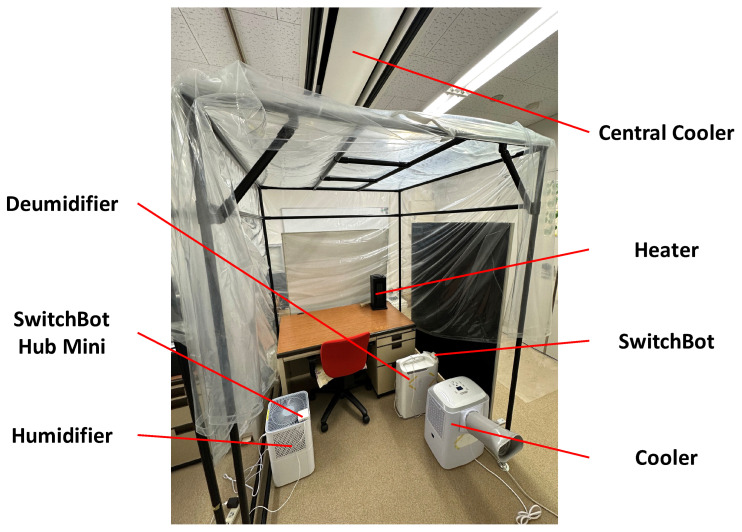
Experimental Environment.

**Figure 5 sensors-23-04965-f005:**
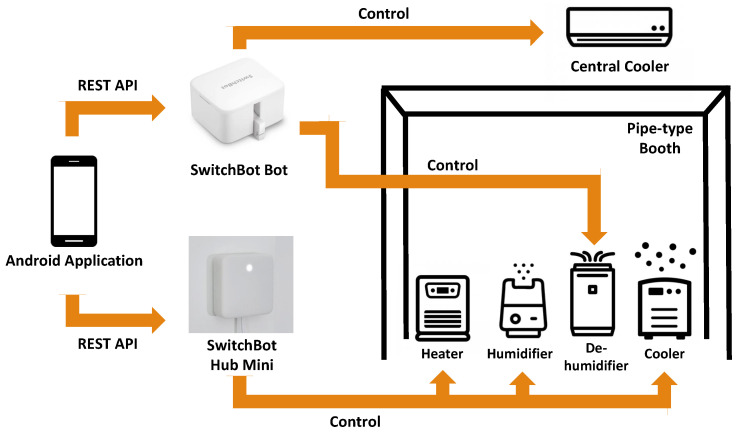
Environmental Control Equipment Diagram.

**Figure 6 sensors-23-04965-f006:**
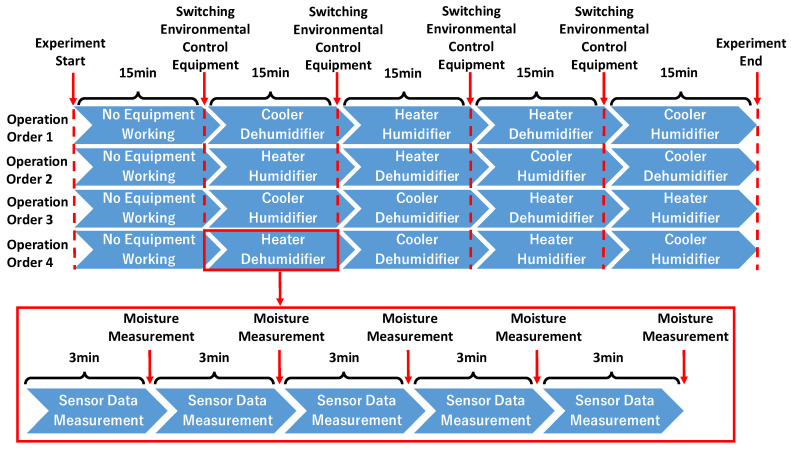
Experimental Procedure Outline.

**Figure 7 sensors-23-04965-f007:**
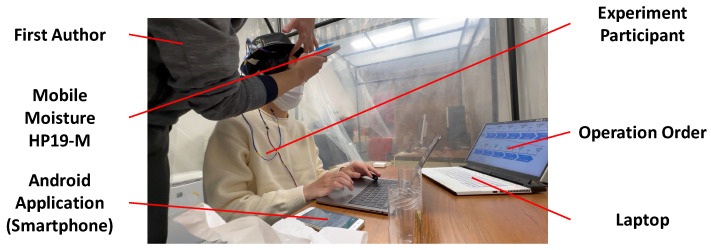
Measurement of Ground Truth in the Experiment.

**Figure 8 sensors-23-04965-f008:**
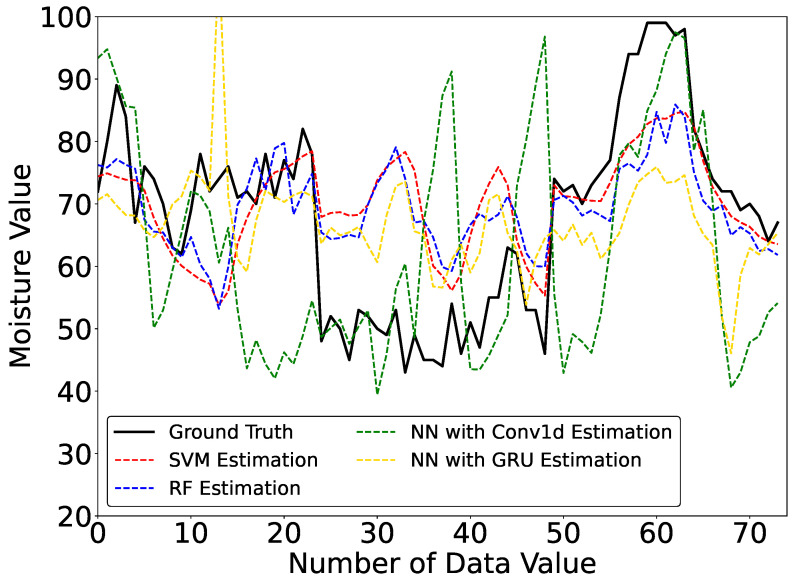
Estimated Curves of Scalp Moisture in the Inter-Subject Evaluation.

**Figure 9 sensors-23-04965-f009:**
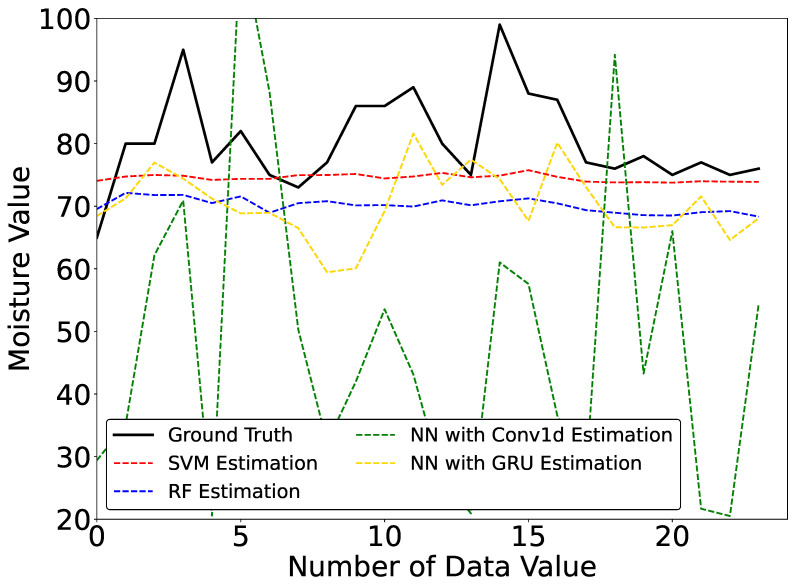
Estimated Curves of Scalp Moisture for One-day Experiment.

**Figure 10 sensors-23-04965-f010:**
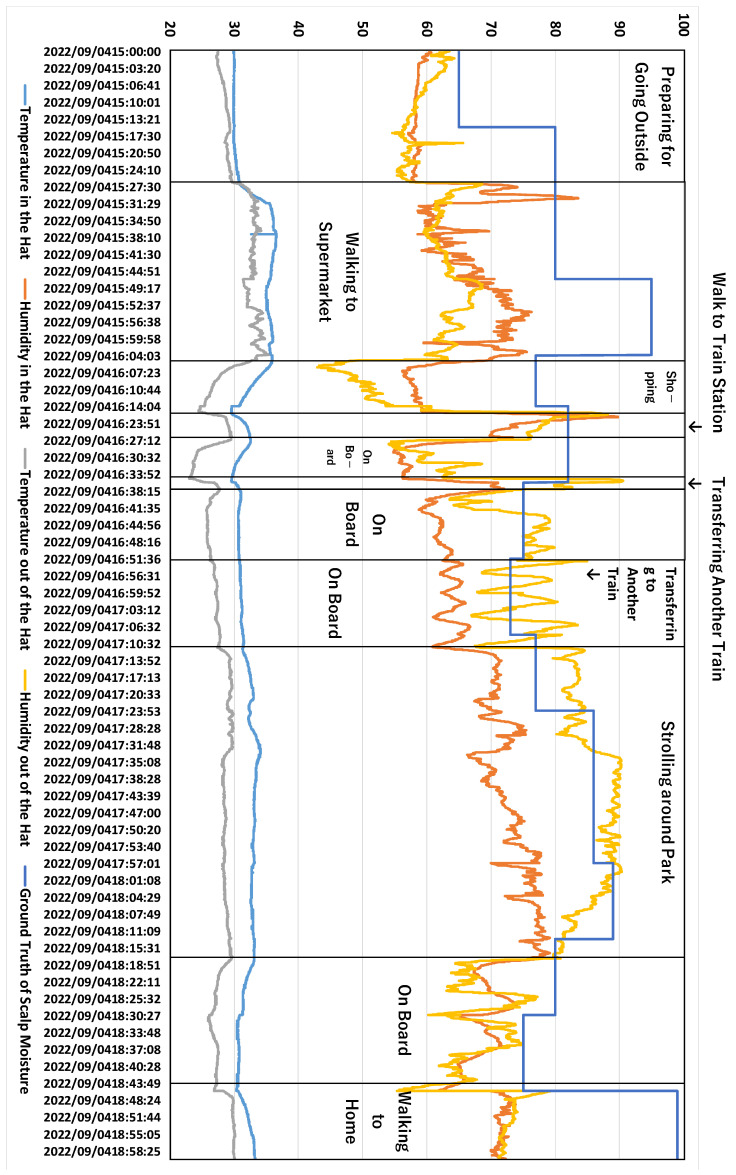
Changes of Environmental Data and Ground Truth of Scalp Moisture in the One-Day Experiment (Part 1).

**Figure 11 sensors-23-04965-f011:**
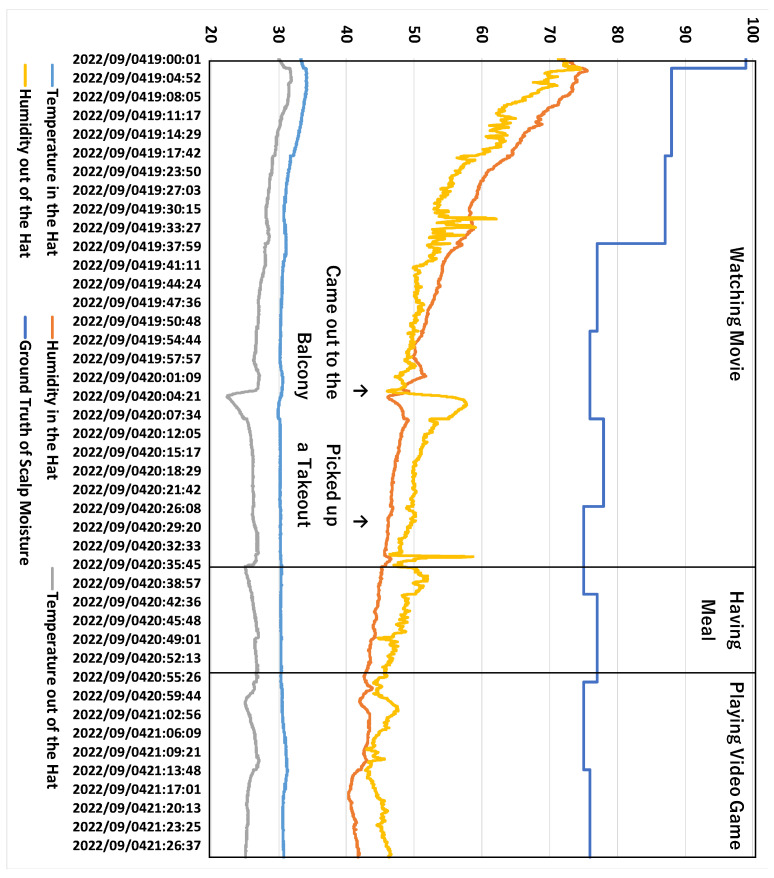
Changes of Environmental Data and Ground Truth of Scalp Moisture in the One-Day Experiment (Part 2).

**Figure 12 sensors-23-04965-f012:**
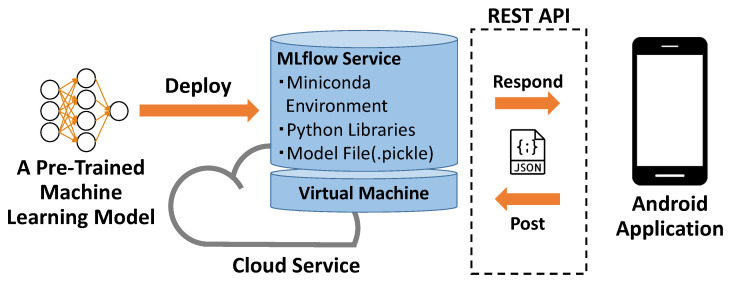
Approach for Applying Estimation System.

**Table 1 sensors-23-04965-t001:** Correspondence between Sensors and Data.

Sensor Types	Used Sensors	Data	Abbreviation
Wearable Sensor	NTC Thermistor	Scalp Surface Temperature	SST
		Core Body Temperature	CBT
	Pulse Sensor	Heartbeats	HB
Environmental Sensor	DHT22	Internal Hat Temperature	IHT
		Internal Hat Humidity	IHH
		External Hat Temperature	EHT
		External Hat Humidity	EHH

**Table 2 sensors-23-04965-t002:** Performance Metrics for Estimation Results in the Inter-Subject Evaluation.

Model	MAE	RMSE
SVM	8.50	10.47
RF	9.75	11.83
NN with Conv1d	14.08	17.76
NN with GRU	9.41	12.26

**Table 3 sensors-23-04965-t003:** Linear Correlation Results of Features of Learning Data against Scalp Moisture.

Feature		Covariance Coefficient	Linear Correlation
CBT	Average of Instantaneous	0.58	Strongly Linearly Correlated
	Average of Mean	0.58	Strongly Linearly Correlated
	Average of Variance	−0.13	Weakly Linearly Correlated
SST	Average of Instantaneous	0.47	Weakly Linearly Correlated
	Average of Mean	0.46	Weakly Linearly Correlated
	Average of Variance	0.18	Weakly Linearly Correlated
IHT	Average of Instantaneous	0.45	Weakly Linearly Correlated
	Average of Mean	0.45	Weakly Linearly Correlated
	Average of Variance	−0.09	Uncorrelated
IHH	Average of Instantaneous	0.26	Weakly Linearly Correlated
	Average of Mean	0.26	Weakly Linearly Correlated
	Average of Variance	0.01	Uncorrelated
EHT	Average of Instantaneous	0.46	Weakly Linearly Correlated
	Average of Mean	0.46	Weakly Linearly Correlated
	Average of Variance	−0.03	Uncorrelated
EHH	Average of Instantaneous	0.00	Uncorrelated
	Average of Mean	0.00	Uncorrelated
	Average of Variance	0.20	Weakly Linearly Correlated
HB	Average of Instantaneous	−0.18	Weakly Linearly Correlated
	Average of Mean	−0.17	Weakly Linearly Correlated
	Average of Variance	−0.03	Uncorrelated

SST: Scalp Surface Temperature, CBT: Core Body Temperature, HB: Heartbeats, IHT: Internal Hat Temperature, IHH: Internal Hat Humidity, EHT: External Hat Temperature, EHH: External Hat Humidity.

**Table 4 sensors-23-04965-t004:** Performance Metrics for Estimation Results in the Intra-Subject Evaluation.

Experiment Participant	SVM		RF		NN with Conv1d		NN with GRU	
	MAE	RMSE	MAE	RMSE	MAE	RMSE	MAE	RMSE
Participant 1	8.11	8.64	1.84	2.43	42.91	43.50	18.19	21.10
Participant 2	13.98	16.49	4.97	6.21	20.92	24.37	38.78	42.97
Participant 3	3.28	3.73	1.78	2.28	29.58	31.49	7.51	9.11
Participant 4	4.73	6.32	2.00	2.89	21.51	25.27	7.61	8.83
Participant 5	4.10	4.53	3.00	3.59	24.72	26.35	7.78	9.43
Participant 6	5.14	5.78	3.00	3.40	28.73	31.87	25.97	28.80
Participant 7	3.32	4.14	2.59	3.35	30.51	30.60	6.22	7.21
Participant 8	10.14	10.30	8.96	9.10	27.80	30.59	30.45	31.66
Participant 9	9.30	11.65	0.96	1.32	38.93	41.00	20.97	27.12
Participant 10	6.25	6.91	3.33	3.98	20.05	21.35	12.33	14.43
Participant 11	7.29	8.22	4.04	5.17	31.58	33.44	37.24	39.34
Participant 12	2.67	3.12	1.71	2.27	19.69	22.48	5.17	7.08
Participant 13	9.46	10.57	3.50	4.68	44.50	46.23	46.61	47.84
Participant 14	4.07	4.72	3.54	4.25	19.44	21.23	26.16	27.84
Participant 15	4.27	4.84	4.10	4.64	23.70	26.62	9.46	11.47
Average	6.41	7.33	3.29	3.97	28.30	30.49	20.03	22.28

**Table 5 sensors-23-04965-t005:** Performance Metrics for Estimation Results in the One-Day Experiment.

Model	MAE	RMSE
SVM	6.74	9.18
RF	10.57	12.31
NN with Conv1d	36.76	39.48
NN with GRU	10.51	12.48

## Data Availability

The research data are not publicly available due to the restrictions from the Ethics Committee of the Graduate School of Engineering, Kobe University.

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
