# Peer review of "Estimating Scalp Moisture in a Hat Using Wearable Sensors"

_sensors, 2023, doi:10.3390/s23104965_

Round 1
Reviewer 1 Report
The paper describes the development of a scalp-based humidity sensor. The work presented is interesting from a device perspective. Although the sensing ideology has been reported extensively, a real-world application-based study is interesting. The paper can be accepted in its current form.
The English language of this manuscript is clear and hence made the manuscript easier to understand.
Author Response
Thanks for your comments. Please see our responses in the attachment.

Reviewer 2 Report
The manuscript reports a hat-shaped device equipped with sensors aiming to estimate scalp moisture content. The topic sounds appropriate for Sensors, whereas it lacks novelty. In my opinion, the Authors can improve their manuscript by addressing a suitable discussion about the most recent advancements achieved by sensor technologies toward smart flexible/wearable systems. The Authors may consider the following articles as a guiding list for the proposed revision:
[1] M. Wang et al. 2022. link: doi.org/10.1038/s41551-022-00916-z
[2] L. M. M. Ferro et al. 2023. link: https://doi.org/10.1002/admt.202300053
[3] X. Wang et al. 2021. link: doi.org/10.1038/s41378-021-00324-4
[4] A. Nawaz et al. 2021. link: https://doi.org/10.1002/adma.202101874[5] X. Yang et al. 2022. link: https://doi.org/10.1002/adma.202201768
Furthermore, please transfer figures 3, 4, 5, 6, and 7 (and their corresponding descriptions) to the supporting information contents.
Moderate/minor editing of the English is required.
Author Response

(The authors gave the same response as above.)

Reviewer 3 Report
1. In the abstract section, it should be indicated the main achievements of this study
2. What is the proposed device in this paper and how does it estimate scalp moisture content?
3. What were the results of the inter-subject evaluation of the proposed device, and which machine learning model achieved the best performance?
4. What are the machine learning models used in the study, and which one has the best performance in both inter-subject and intra-subject evaluations?
5. Discussed more about the machine learning model in introduction section.
6. Which machine learning model had the smallest MAE in the inter-subject evaluation and which had the smallest average MAE in the intra-subject evaluation?
7. What is the authors' assessment of the results, and what do they suggest as future directions for improving scalp moisture estimation accuracy?
8. What is the cloud-service-based method presented in the paper for applying the scalp moisture estimation results?
Extensive editing of English language required
Author Response

(The authors gave the same response as above.)

Round 2
Reviewer 2 Report
The Authors improved the discussions as suggested by the Reviewer. My recommendation is accepted as it is.
I would say that small editing of the English is required.
Reviewer 3 Report
Accept in present form
Moderate editing of english